# Antigenic evolution of human influenza H3N2 neuraminidase is constrained by charge balancing

Yiquan Wang[1†], Ruipeng Lei[1†], Armita Nourmohammad[2,3,4], Nicholas C Wu[1,5,6,7]*

[1]Department of Biochemistry, University of Illinois at Urbana-Champaign, Urbana, United States; [2]Department of Physics, University of Washington, Seattle, United States; [3]Max Planck Institute for Dynamics and Self-Organization, Göttingen, Germany; [4]Fred Hutchinson Cancer Research Center, Seattle, United States; [5]Center for Biophysics and Quantitative Biology, University of Illinois at Urbana-Champaign, Urbana, United States; [6]Carl R. Woese Institute for Genomic Biology, University of Illinois at Urbana-Champaign, Urbana, United States; [7]Carle Illinois College of Medicine, University of Illinois at Urbana-Champaign, Urbana, United States

**Abstract** As one of the main influenza antigens, neuraminidase (NA) in H3N2 virus has evolved extensively for more than 50 years due to continuous immune pressure. While NA has recently emerged as an effective vaccine target, biophysical constraints on the antigenic evolution of NA remain largely elusive. Here, we apply combinatorial mutagenesis and next-generation sequencing to characterize the local fitness landscape in an antigenic region of NA in six different human H3N2 strains that were isolated around 10 years apart. The local fitness landscape correlates well among strains and the pairwise epistasis is highly conserved. Our analysis further demonstrates that local net charge governs the pairwise epistasis in this antigenic region. In addition, we show that residue coevolution in this antigenic region is correlated with the pairwise epistasis between charge states. Overall, this study demonstrates the importance of quantifying epistasis and the underlying biophysical constraint for building a model of influenza evolution.

**\*For correspondence:**
nicwu@illinois.edu

[†]These authors contributed equally to this work

## Editor's evaluation

This paper presents a systematic analysis of the fitness landscape of the influenza virus protein neuraminidase (NA). The authors generate 864 different combinations of amino acids at seven positions in six genetic backgrounds sampled 10 years apart and measure the fitness of the resulting virus. This fitness landscape is characterized by strong epistatic interactions, including a strong tendency to maintain the local charge of the protein. Such systematic characterizations of important proteins of viral pathogens are crucial to develop principled models to understand and predict their evolution.

## Introduction

There are two major antigens on the surface of influenza virus, hemagglutinin (HA) and neuraminidase (NA). Although influenza vaccine development has traditionally focused on HA, NA has emerged as an effective vaccine target in the past few years because recent studies have shown that NA immunity has a significant role in protection against influenza infection (*Monto et al., 2015*; *Weiss et al., 2020*; *Couch et al., 2013*; *Memoli et al., 2016*; *Krammer et al., 2018*). Influenza NA has an N-terminal transmembrane domain, a stalk domain, and a C-terminal head domain. The head domain of NA

functions as an enzyme to cleave the host receptor (i.e., sialylated glycan), which is essential for virus release. Most NA antibodies target the surface loop regions that surround the highly conserved catalytic active site (*Gulati et al., 2002*; *Malby et al., 1994*; *Venkatramani et al., 2006*; *Colman et al., 1983*; *Air, 2012*; *Zhu et al., 2019*; *Tulip et al., 1992*). Due to the need to constantly escape from herd immunity (also known as antigenic drift), both HA and NA of human influenza virus have evolved extensively (*Kilbourne et al., 1990*; *Sandbulte et al., 2011*; *Westgeest et al., 2015*). For example, since influenza H3N2 virus entered the human population in 1968, its HA and NA have accumulated more than 83 and 73 amino acid mutations, respectively (*Figure 1—figure supplement 1*), which accounted for ~15% of their protein sequences. However, the evolution of NA is much less well characterized as compared to HA.

To understand how the evolutionary trajectories of NA are being shaped, it is important to characterize the underlying biophysical constraints that govern the fitness of individual amino acid mutations and epistatic interactions between mutations (*Starr and Thornton, 2016*; *Echave and Wilke, 2017*; *Gong et al., 2013*). Epistasis is a phenomenon in which the fitness effect of a mutation is dependent on the presence or absence of other mutations. Since epistasis can lead to differential fitness effects of a given mutation on different genetic backgrounds, it can restrict evolutionary trajectories or open up a new functional sequence space that would otherwise be inaccessible (*Wu et al., 2016*). As a result, epistasis is a primary challenge for predicting evolution (*Miton and Tokuriki, 2016*; *Luksza and Lässig, 2014*). Nevertheless, epistasis is pervasive in natural evolution in general (*Breen et al., 2012*) and has been shown to influence the evolution of influenza virus (*Gong et al., 2013*; *Wu et al., 2016*; *Lyons and Lauring, 2018*; *Wu et al., 2018*; *Kryazhimskiy et al., 2011*; *Wu et al., 2020*; *Koel et al., 2019*; *Bloom et al., 2010*; *Abed et al., 2011*; *Wu et al., 2013*; *Duan et al., 2014*). While epistasis is critical for the emergence of oseltamivir-resistant mutants of influenza NA (*Bloom et al., 2010*; *Abed et al., 2011*; *Wu et al., 2013*; *Duan et al., 2014*), the role of epistasis and the underlying biophysical constraints on NA antigenic evolution remains largely unclear.

Deep mutational scanning combines saturation mutagenesis and next-generation sequencing to determine the phenotypic effects of numerous mutations in a highly parallel manner (*Lee et al., 2018*; *Narayanan and Procko, 2021*; *Fowler and Fields, 2014*). Deep mutational scanning has been employed to measure the replication fitness effect of all possible single amino acid mutations across different influenza proteins (*Lee et al., 2018*; *Thyagarajan and Bloom, 2014*; *Doud et al., 2015*; *Soh et al., 2019*; *Hom et al., 2019*), which in turn can help to model the natural evolution of human influenza virus (*Lee et al., 2018*; *Thyagarajan and Bloom, 2014*). However, most deep mutational scanning experiments lack the power to systematically measure the fitness of variants with two or more mutations, hence epistasis. More recently, combinatorial mutagenesis was used in conjunction with next-generation sequencing to examine the local fitness landscape and to identify epistasis in influenza HA (*Wu et al., 2018*; *Wu et al., 2020*; *Wu et al., 2017*). Unlike saturation mutagenesis in conventional deep mutational scanning, combinatorial mutagenesis enables us to measure the fitness of high-order mutants. By dissecting the mechanistic basis of epistasis, these studies have provided important insight into the biophysical constraints on HA antigenic evolution (*Wu et al., 2018*; *Wu et al., 2020*).

In this study, we aim to understand how epistatic effects influence NA antigenic evolution and investigate the underlying biophysical constraints. Specifically, we coupled combinatorial mutagenesis and next-generation sequencing to characterize the local fitness landscape of an NA antigenic region in six different human H3N2 strains that were isolated ~10 years apart. Our results indicate that the local fitness landscape of this NA antigenic region is highly correlated across six different genetic backgrounds. In-depth analyses further demonstrate that local net charge balancing is a biophysical constraint that governs the epistasis within this NA antigenic region. Lastly, we show that epistasis is correlated with residue coevolution in naturally circulating influenza strains.

## Results

### Local fitness landscape of an antigenic region on NA

When the structure of N2 NA was first reported in 1983 (*Varghese et al., 1983*), seven regions (I–VII) were proposed to be targeted by antibodies (*Colman et al., 1983*). Within regions I–III, mutations at residues 329, 344, 368, and 370 have been shown to escape monoclonal antibodies (*Gulati et al.,*

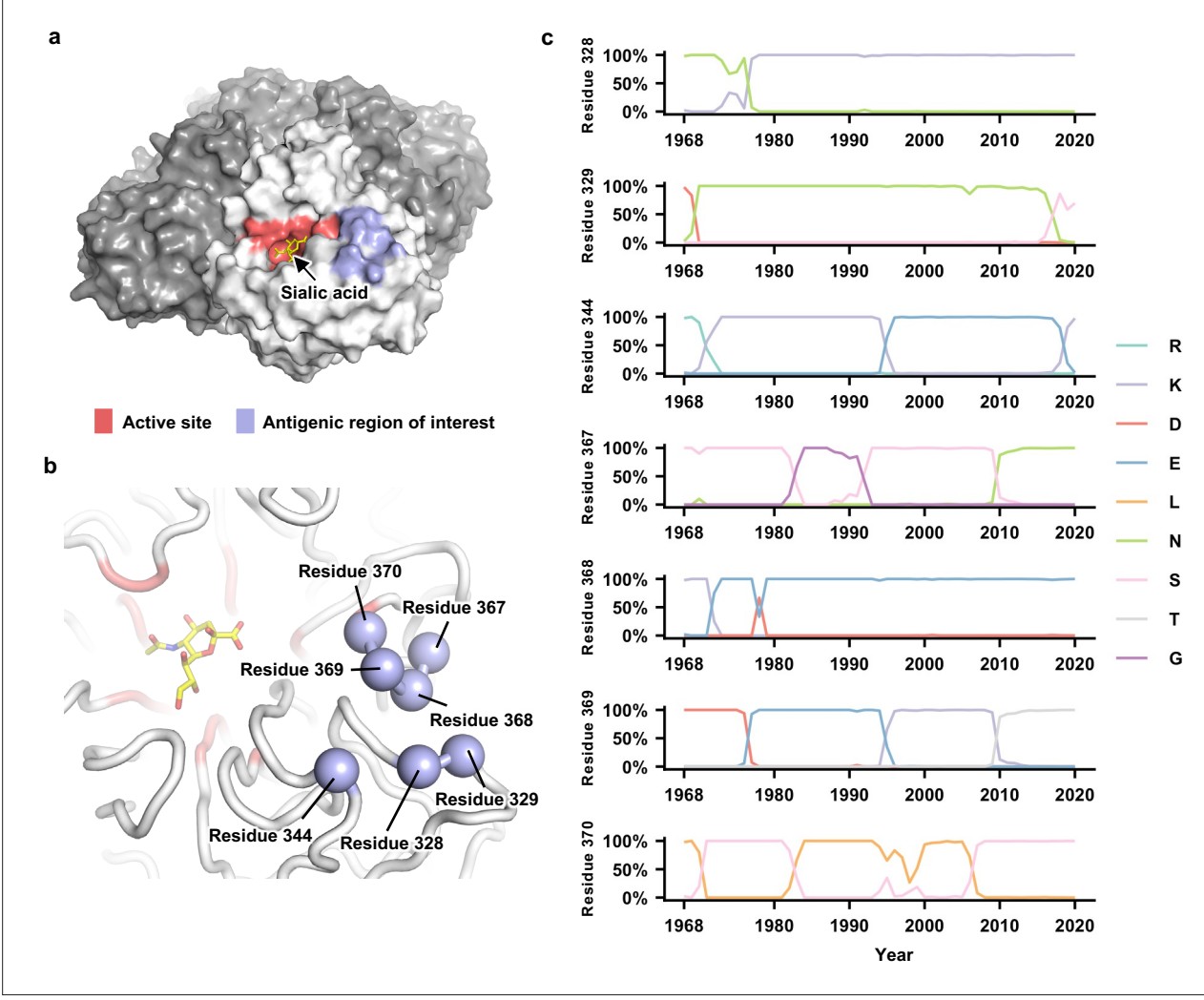

**Figure 1.** Natural evolution of an antigenic region in human H3N2 neuraminidase (NA). (**a**) The enzyme active site and the antigenic region of interest are highlighted as red and blue, respectively, on one protomer of the NA tetramer. The sialic acid within the active site is shown in yellow. (**b**) Seven residues in the antigenic region of interest are highlighted in blue spheres. (**c**) Natural occurrence frequencies of the amino acid variants that have a natural occurrence at >50% in any given year at the residues of interest are shown (**Figure 1—source data 1** and **Figure 1—source data 2**). Of note, only those amino acid variants with a natural occurrence at >80% in any given year were included in our mutant libraries. Therefore, although D368 is shown in this plot, it was not included in our mutant libraries since it only reached a maximum occurrence of 67%.

The online version of this article includes the following figure supplement(s) for figure 1:

**Source data 1.** Human H3N2 neuraminidase (NA) protein sequences.

**Source data 2.** Natural occurrence frequencies of the major amino acid variants at the residues of interest in different years.

**Figure supplement 1.** Accumulation of amino acid mutations in the hemagglutinin (HA) and neuraminidase (NA) of human H3N2 viruses.

**Figure supplement 2** The frequency of haplotypes at the neuraminidase (NA) antigenic region of interest in circulating human H3N2 strains.

**Figure supplement 3.** Correlations of fitness measurements between replicates.

*2002*; *Varghese et al., 1988*; *Air et al., 1985*). These four residues, along with residues 328 (region I), 367 (region III), and 369 (region III), form a cluster of seven residues that are very close in space (*Figure 1a and b*). Both residues 328 and 367 are under positive selection in human H3N2 NA (*Westgeest et al., 2012*), whereas coevolution of residues 367 and 369 in human H3N2 NA has created an N-glycosylation site (NXT) at residue 367 during 2010 (*Figure 1c*). These observations indicate that residues 328, 367, and 369 also participated in the antigenic drift of human H3N2 NA. By focusing on this seven-residue antigenic region, this study aimed to dissect the biophysical constraints on NA antigenic evolution.

We first compiled a list of amino acid variants at NA residues 328, 329, 344, 367, 368, 369, and 370 that reached an occurrence frequency of >80% in any given year during the natural evolution of human H3N2 viruses since 1968 (*Figure 1c*). This list includes two amino acid variants at residue 328 (Asn and Lys), three at residue 329 (Asn, Ser, and Asp), three at residue 344 (Lys, Glu, and Arg), three at residue 367 (Asn, Ser, and Gly), two at residue 368 (Lys and Glu), four at residue 369 (Lys, Glu, Asp, and Thr), and two at residue 370 (Leu and Ser). Together, there are 2 × 3 × 3 × 3 × 2 × 4 × 2 = 864 possible amino acid combinations (also called haplotypes) across these seven residues, although only 53 of them have been observed in naturally circulating human H3N2 strains (*Figure 1—figure supplement 2*). Using PCR primers that carry degenerate nucleotides (see Materials and methods and *Supplementary file 1*), these 864 variants were introduced into the NA of six different strains (genetic backgrounds) from 1968 to 2019, namely, A/Hong Kong/1/1968 (HK68), A/Bangkok/1/1979 (Bk79), A/Beijing/353/1989 (Bei89), A/Moscow/10/1999 (Mos99), A/Victoria/361/2011 (Vic11), and A/Hong Kong/2671/2019 (HK19). All these six strains were historical vaccine strains and isolated approximately 10 years apart.

To measure the virus replication fitness of all 864 variants in the six different genetic backgrounds of interest (i.e., HK68, Bk79, Bei89, Mos99, Vic11, and HK19), we employed a high-throughput experimental approach that coupled combinatorial mutagenesis and next-generation sequencing. Unlike conventional deep mutational scanning, which studies all possible single amino acid mutations across a protein or domain of interest, our approach analyzes all possible combinations of a subset of mutations. Subsequently, six different local fitness landscapes, each with 864 variants, were obtained. Fitness value of each variant, which was defined in log scale (see Materials and methods), was normalized to the corresponding wild type (WT), such that the WT fitness value was 0, whereas positive and negative fitness values represented beneficial and deleterious variants, respectively. Two biological replicates of each experiment were performed. Overall, a high correlation was observed between the replicates (Pearson correlation = 0.81–0.89) except for HK68 and Bk79 (0.39 and 0.63, respectively), mostly due to the high measurement noise for low fitness variants (*Figure 1—figure supplement 3*).

## Dynamics of the local fitness landscape across genetic backgrounds

To examine whether the local fitness landscape of the NA antigenic region of interest changes over time, fitness measurements of the 864 variants were compared across different genetic backgrounds (*Figure 2a and b*). Interestingly, while most variants were strongly deleterious (i.e., had a fitness of <-1) in HK68, variants with a fitness of <-1 were rare in other genetic backgrounds (*Figure 2a*). Notably, the difference in variant fitness distribution across genetic backgrounds was not due to the difference in their WT replication fitness since the viral titers from a rescue experiment were similar among HK68 WT, Bei89 WT, and Mos99 WT (*Figure 2—figure supplement 1*). In contrast, the fitness of individual variants correlated well across genetic backgrounds (Pearson correlation = 0.48–0.79, *Figure 2b*), despite their differences in variant absolute fitness values (*Figure 2a*). These results demonstrate that although epistasis exists between the seven-residue antigenic region and the rest of the NA sequence, such epistasis is largely variant-nonspecific. Consequently, the topology of the local fitness landscape is highly conserved across genetic backgrounds. This observation is very different from a similar study on a major antigenic site of HA, where the topology of the local fitness landscape differed dramatically among genetic backgrounds (*Wu et al., 2020*).

We further examined the fitness of naturally occurring variants in different genetic backgrounds (HK68, Bk79, Bei89, Mos99, Vic11, and HK19) (*Figure 2c*). Most natural variants have a fitness between –0.5 and 0.5 on the background of Bk79, Bei89, Mos99, Vic11, and HK19. In contrast, most natural variants from 1980s onward are strongly deleterious with fitness <-1 on the background of HK68, indicating many natural variants would not have emerged if the genetic background had not evolved. In other words, the emergence of natural variants in the antigenic region of interest is contingent on the evolution of the rest of the NA sequence. This result is consistent with the observation that HK68 has a much lower mutational tolerance at this antigenic region (*Figure 2a*).

## Conservation of pairwise epistasis across genetic backgrounds

To understand the biophysical constraints on NA antigenic evolution, the local fitness landscapes were decomposed into additive fitness effects of individual amino acid variants and pairwise epistasis between amino acid variants (*Starr and Thornton, 2016*). Additive fitness describes the independent

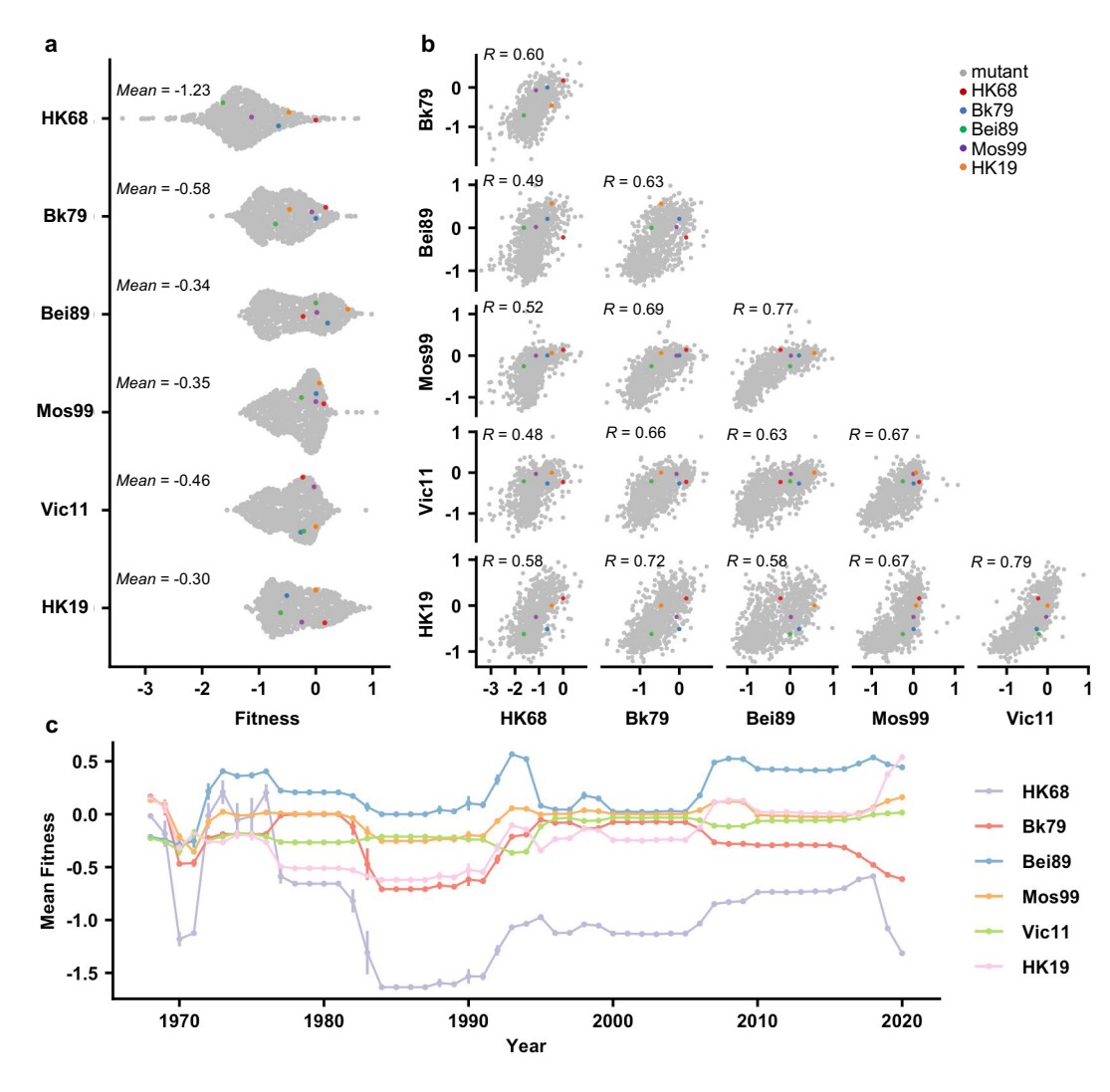

**Figure 2.** Comparing the local fitness landscapes of the neuraminidase (NA) antigenic region in six human H3N2 strains. (**a**) Variant fitness distributions in different genetic backgrounds are shown as a sina plot (***Figure 2—source data 1***). Each data point represents one variant. A total of 864 data points are present for each row (each genetic background). (**b**) Correlations of variant fitness distribution among different genetic backgrounds are shown, with each data point representing one variant. Pearson correlation coefficients (R) are indicated. (**a, b**) Data points corresponding to the wild type (WT) sequences of HK68, Bk79, Bei89, Mos99, and HK19 are colored as indicated. Of note, the WT sequence of Vic11 contains a naturally rare variant T329. Therefore, the WT sequence of Vic11 was not included in our mutant libraries. (**c**) Naturally occurring variants were grouped by the year of isolation, and their mean fitness in different genetic backgrounds is shown (***Figure 2—source data 2***). Different genetic backgrounds are represented by different lines, which are color coded as indicated on the right. Error bars represent the standard error of mean. This analysis included 66,562 NA sequences from human H3N2 strains that were isolated between 1968 and 2020 (***Figure 1—source data 1***).

The online version of this article includes the following figure supplement(s) for figure 2:

**Source data 1.** Fitness value of each variant across six different genetic background.

**Source data 2.** The yearly mean fitness of naturally occurring variants in different genetic backgrounds.

**Figure supplement 1.** Virus rescue experiment of wild type (WT) strains.

contributions of each amino acid variant to fitness, whereas pairwise epistasis describes the nonadditive interactions between amino acid variants. For each genetic background, additive fitness and pairwise epistasis were inferred from the variant fitness data using an established statistical learning model (***Tareen et al., 2020***) (see Materials and methods). The model was evaluated using repeated k-fold cross-validation and hyperparameters were chosen by maximizing the $R^2$ of model prediction and the Pearson correlation coefficient of model parameters (***Figure 3—figure supplement 1***). While

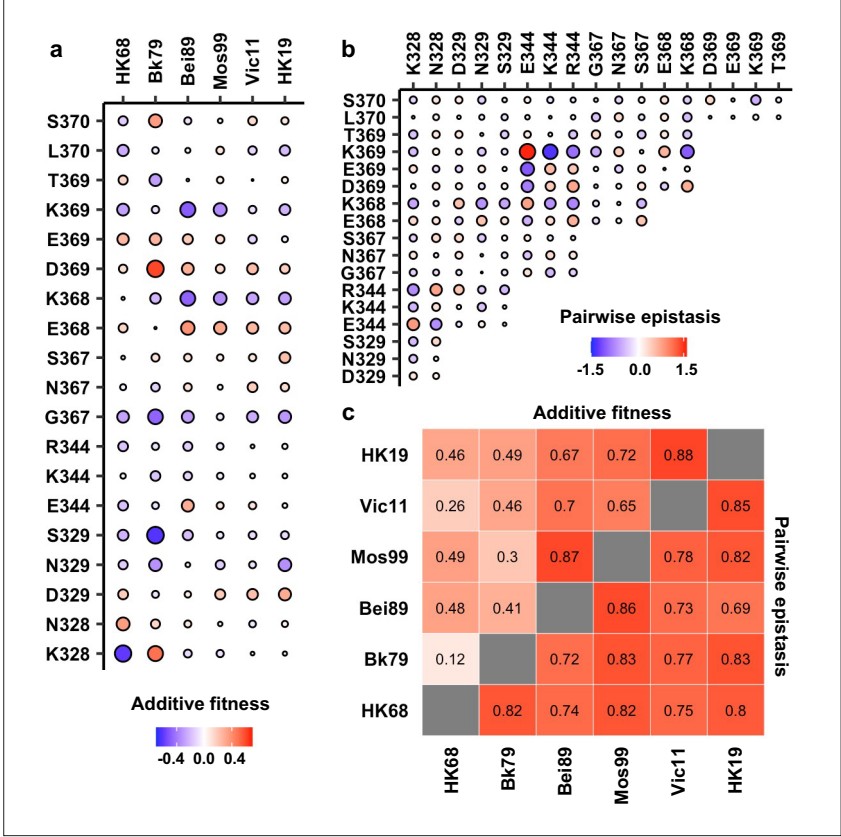

**Figure 3.** Inference of additive fitness and pairwise epistasis. (**a**) Parameters for additive fitness in different genetic backgrounds are shown. (**b**) Pairwise epistatic effects show strong correlations among six different genetic backgrounds, and therefore, Bk79 is shown as a representative. The identity of a given double amino acid variant is represented by the labels on the x- and y-axes. The same plots for other genetic backgrounds are shown in *Figure 3—figure supplement 5*. (**a, b**) Positive additive fitness and pairwise epistasis are in red. Negative additive fitness and pairwise epistasis are in blue. The magnitude is proportional to the size of the circle. (**c**) Correlation matrices of additive fitness and pairwise epistasis among six genetic backgrounds are shown as a heatmap. See *Figure 3—figure supplements 4 and 5* for the related scatter plots.

The online version of this article includes the following figure supplement(s) for figure 3:

**Figure supplement 1.** Evaluation of model hyperparameters using repeated k-fold cross-validation.

**Figure supplement 2.** Correlations of additive fitness among genetic backgrounds.

**Figure supplement 3.** Correlations of pairwise epistasis among genetic backgrounds.

**Figure supplement 4.** Evaluation of additive-only model.

**Figure supplement 5.** Pairwise epistasis in different genetic backgrounds.

the correlations between additive fitness contributions varied hugely across genetic backgrounds (Pearson correlation = 0.12–0.88, *Figure 3a and c*, *Figure 3—figure supplement 2*), we observed generally strong correlations between pairwise epistatic effects across the six different genetic backgrounds (Pearson correlation = 0.69–0.86, *Figure 3b and c*, *Figure 3—figure supplement 3*). Of note, the variation of additive fitness contributions across genetic backgrounds does not seem to strictly depend on the similarity between genetic backgrounds since the correlation between the additive fitness contributions of HK68 and HK19 (Pearson correlation = 0.46) is much higher than that of HK68 and Bk79 (Pearson correlation = 0.12), which have a shorter time separation. This observation points at a possibility for complex epistatic interactions between the antigenic region of interest and other regions on NA. Overall, these results indicate that pairwise epistasis, but not additive fitness, is highly conserved at the NA antigenic region of interest across genetic backgrounds. Consistently, an 'additive-only' model, without accounting for epistasis, shows a poor fit to the fitness landscape data compared to the model above with epistasis (*Figure 3—figure supplement 4a*), despite the fact

that the inferred additive fitness effects correlate well between the two models (*Figure 3—figure supplement 4b*).

## Local net charge imposes a biophysical constraint on NA antigenic evolution

Next, we investigated the biophysical constraints that have led to such conserved patterns of epistatic interactions in the NA antigenic region of interest. We noticed that amino acid variants with opposite charges usually exhibited positive epistasis, whereas amino acid variants with the same charge usually exhibited negative epistasis (*Figures 3b and 4a*, *Figure 3—figure supplement 5*, *Figure 4—figure supplement 1*; see Materials and methods). In contrast, pairwise interactions that involved a neutral amino acid variant did not show any bias towards positive or negative epistasis. These results suggest that balancing of charged amino acid variants imposes a key biophysical constraint on the evolution of this NA antigenic region.

To further probe the mechanism of epistasis in this NA antigenic region, we analyzed a published crystal structure of human H3N2 NA that has K328, E344, and K369 (*Figure 4b*; *Venkatramani et al., 2006*). Both variant pairs E344/K369 and K328/E344 exhibited positive epistasis across all genetic backgrounds, whereas K328/K369 exhibited negative epistasis (*Figure 3—figure supplement 5*). In fact, E344 and K369 had the strongest positive epistasis in five of the six genetic backgrounds of interest (except Bei89). Our structural analysis showed that the distance between the side-chain carboxylate oxygen of E344 and the side-chain amine nitrogen of K369 is 6.6 Å, whereas the distance between the side-chain carboxylate oxygen of E344 and the side-chain amine nitrogen of K328 is 5.5 Å (*Figure 4b*, *Figure 4—figure supplement 2*). At these distances, salt bridges cannot be formed and the electrostatic attraction force is negligible (*Kumar and Nussinov, 2002*; *Shashikala et al., 2019*). Similarly, the distance between the side-chain amine nitrogen atoms of K328 and K369 is 11.6 Å (*Figure 4b*), which is too far for any significant electrostatic repulsion force (*Shashikala et al., 2019*). As a result, direct side-chain–side-chain interaction via electrostatic attraction or repulsion is unlikely to be a major determinant for epistasis in this NA antigenic region. Consistently, pairwise epistasis between two amino acid variants does not correlate with their side-chain–side-chain distances (*Figure 4d–f*, *Figure 4—figure supplement 3*), further substantiating that direct side-chain–side-chain interaction is not a determinant for epistasis here.

We then analyzed the relationship between variant fitness and local net charge as a molecular phenotype. Here, local net charge was defined as the sum of charges at the seven residues of interest (residues 328, 329, 344, 367, 368, 369, and 370), where positively charged amino acids (K and R) were +1 and negatively charged residues (D and E) –1. In all genetic backgrounds, variants tended to have a higher fitness when the local net charge was around –1, while variants with a more positive or negative local net charge usually had a lower fitness (*Figure 4c*, *Figure 4—figure supplement 4*). Overall, our results demonstrate that the local net charge is a key molecular phenotype with a biophysical function that is under balancing selection. This biophysical phenotype imposes a strong constraint on the evolution of the NA antigenic region, reflected in the conserved epistatic interactions between amino acid variants of this region.

## Impact of local net charge and epistasis on NA evolution

Next, we aimed to understand whether the local net charge at the NA antigenic region of interest influences its evolution in circulating human H3N2 strains. We retrieved 66,562 human H3N2 NA sequences spanning from 1968 to 2020 from the Global Initiative for Sharing Avian Influenza Data (GISAID) (*Shu and McCauley, 2017*). Most natural variants in the antigenic region of interest had a local net charge between –1 and +1 (*Figure 5a*). Natural variants with a local net charge of –3,–2, +2, or +3 could also be observed but were rare. This observation suggests that the natural evolution of this NA antigenic region is constrained by balancing the local net charge.

We further tested whether epistasis is correlated with the evolution of local net charge at the NA antigenic region of interest. A coarse-grained analysis that only considered three charged states, namely, positive, negative, and neutral, for each residue was performed (see Materials and methods, *Figure 5—figure supplement 1*). In this analysis, amino acids were classified into (+) as positively charged (amino acids K/R), (-) as negatively charged (amino acids D/E), and (n) as neutral representing the remaining amino acids. We then computed the pairwise epistasis between different charge classes

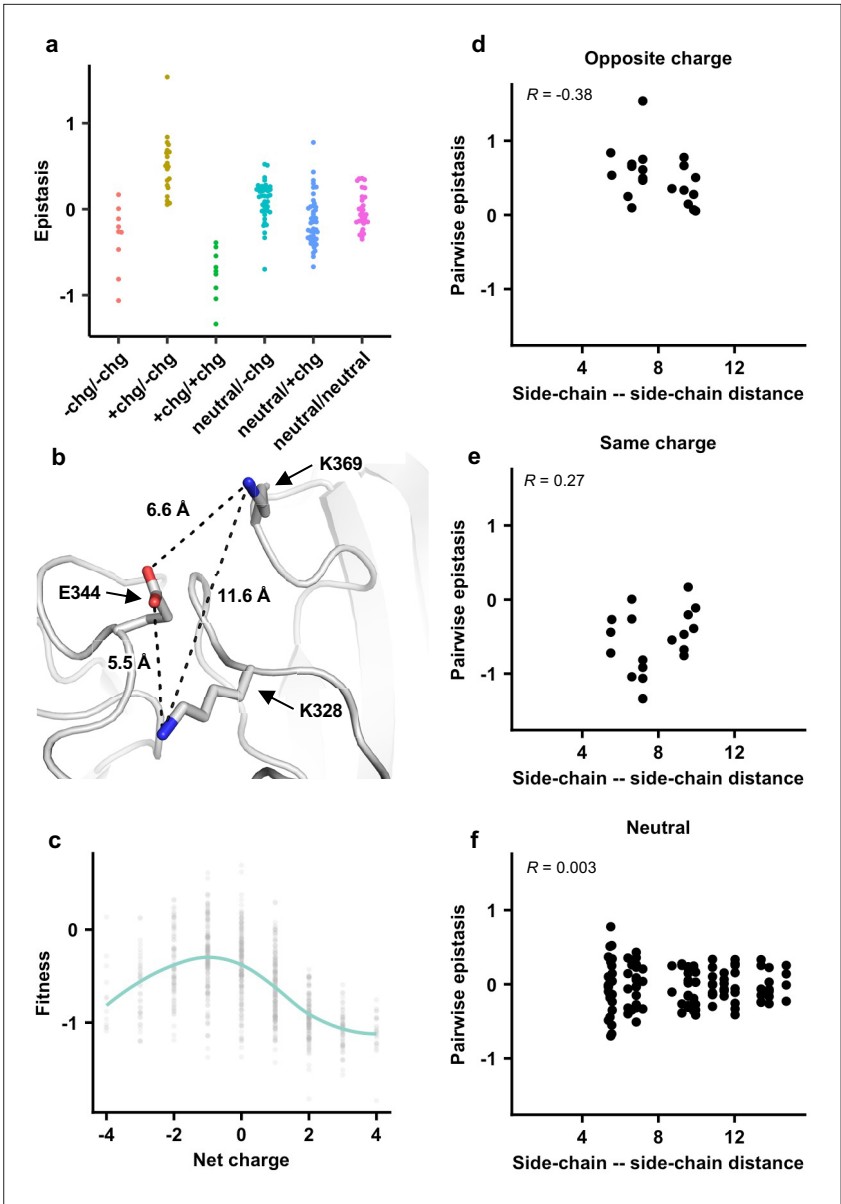

**Figure 4.** The importance of local net charge in the neuraminidase (NA) antigenic region. (**a**) Pairwise epistasis in genetic background Bk79 is shown and classified based on amino acid charges. +chg represents positively charged amino acids (K/R), -chg represents negatively charged amino acids (D/E), and neutral represents the remaining amino acids. The same plots for other genetic backgrounds are shown in *Figure 4—figure supplement 1*. (**b**) NA structure from H3N2 A/Memphis/31/98 (PDB 2AEP), which has K328, E344, and K369, was analyzed (*Gulati et al., 2002*). A similar NA structure from H3N2 A/Tanzania/205/2010, which has K328 and E344, is shown in *Figure 4—figure supplement 2*. (**c**) The relationship between variant fitness and net charge in genetic background Bk79 is shown. A smooth curve was fitted by loess and shown in teal. The same plots for other genetic backgrounds are shown in *Figure 4—figure supplement 4*. (**d–f**) Relationship between the side-chain–side-chain distances (*Figure 4—source data 1*) and epistasis for (**d**) variant pairs with opposite charges, (**e**) variant pairs with the same charge, and (**f**) variant pairs that involve a neutral amino acid in genetic background Bk79 is shown. The same plots for other genetic backgrounds are shown in *Figure 4—figure supplement 3*. Pearson correlation coefficient (R) is indicated.

The online version of this article includes the following figure supplement(s) for figure 4:

**Source data 1.** Pairwise side-chain–side-chain distances within neuraminidase (NA) antigenic region.

**Figure supplement 1.** Pairwise epistasis was classified based on the side-chain charge in each pair of amino acid variants.

*Figure 4 continued on next page*

*Figure 4 continued*

**Figure supplement 2.** Neuraminidase (NA) structure from H3N2 A/Tanzania/205/2010 (PDB 4GZO) (*Zhu et al., 2012*), which has K328 and E344, is shown.

**Figure supplement 3.** Relationship between the side-chain–side-chain distance and pairwise epistasis.

**Figure supplement 4.** Relationship between variant fitness and the local net charge.

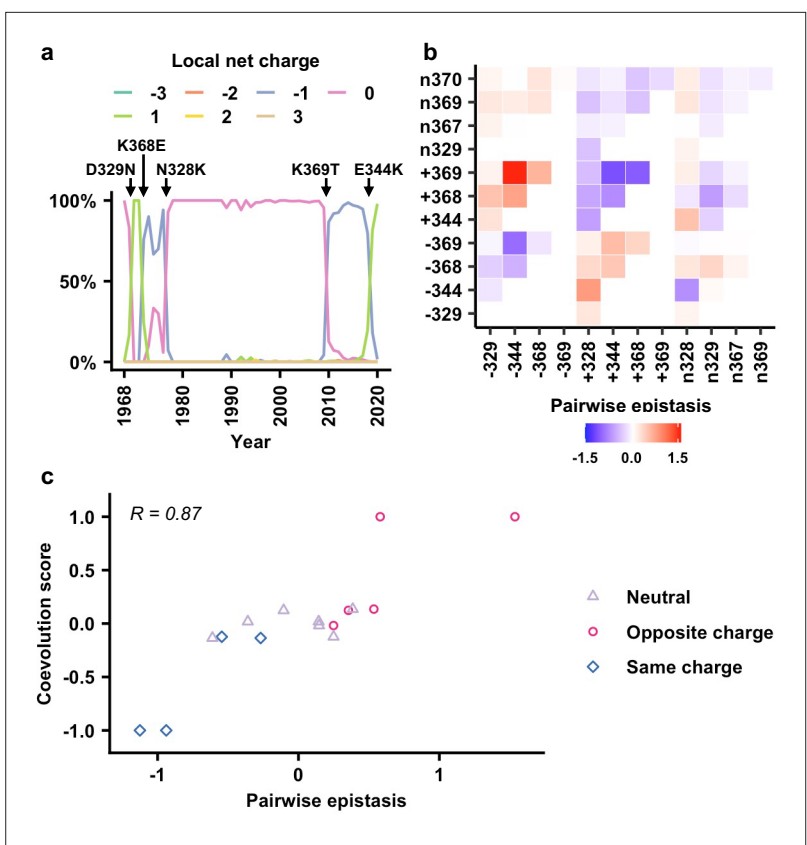

**Figure 5.** Correlation between epistasis and coevolution of charge states in the neuraminidase (NA) antigenic region. (**a**) The natural evolution of frequency of haplotypes by local net charge at the NA antigenic region of interest is shown (*Figure 5—source data 1*). Key mutation events that led to changes in local net charge are indicated. (**b**) Pairwise epistasis of charge states among different residues. Amino acids were classified based on charges. (+) represents positively charged amino acids (K/R). (-) represents negatively charged amino acids (D/E). (n) represents the remaining amino acids. (**c**) Relationship between coevolution score (*Figure 5—source data 2*) and pairwise epistasis in genetic background Bk79 is shown. Pearson correlation coefficient (R) is indicated. The same plots for other genetic backgrounds are shown in *Figure 5—figure supplement 4*.

The online version of this article includes the following figure supplement(s) for figure 5:

**Source data 1.** The local net charge at the neuraminidase (NA) antigenic region of interest in different strains from 1968 to 2020.

**Source data 2.** Pairwise coevolution score for pairs of charge states that emerge or disappear within 5 years.

**Figure supplement 1.** The natural occurrence frequencies of the major amino acid charge states are shown.

**Figure supplement 2.** Pairwise epistasis of charged states among seven residues.

**Figure supplement 3.** Schematic overview of calculating the coevolution score.

**Figure supplement 4.** Correlations between coevolution score and pairwise epistasis in different genetic backgrounds.

**Figure supplement 5.** An ensemble of evolutionary trajectories constructed from fitness data.

**Figure supplement 6.** The natural occurrence frequencies of the charge states in the homologous regions of H1N1 and influenza B are shown.

in two loci by averaging over all the corresponding amino acids in each class (*Figure 5b*, *Figure 5—figure supplement 2*). For example, the epistasis between positive charge at residue 344 and negative charge at residue 369 (i.e., +344/–369) is the averaged epistasis over K344/D369, R344/D369, K344/E369, and R344/E369. In addition, we evaluated a coevolution score between a pair of charged states at two different residues (Materials and methods, *Figure 5—figure supplement 3*). In our definition, two charge states that emerged or disappeared shortly one after another would have a positive coevolution score. In contrast, emergence of a charge state in one residue followed by disappearance of a charge state in another residue would result in a negative coevolution score. Our analysis only included pairs of charge states that emerged or disappeared within 5 years from each other. Since residues 367 and 370 were dominated by neutral charge since 1968, they were not included on our coevolution analysis (*Figure 5—figure supplement 1*).

When we compared the coevolution score with the pairwise epistasis (*Figure 5c*, *Figure 5—figure supplement 4*), high correlation was observed (Pearson correlation = 0.79–0.92). Specifically, pairs with opposite charges usually have positive coevolution scores and positive epistasis, whereas pairs with the same charge usually have negative coevolution scores and negative epistasis. We also attempted to use the fitness data to predict the identity of amino acid mutations along the evolutionary trajectory. Nonetheless, the realized evolutionary trajectories in nature are seldom the most probable ones among an ensemble of trajectories that were constructed from the fitness data (*Figure 5—figure supplement 5*). As a result, while our analysis demonstrates that a biophysically grounded epistatic landscape is correlated with the natural coevolution of residues in this NA antigenic region, additional considerations are needed for the prediction of the exact amino acid mutations along the evolutionary trajectory.

## Discussion

Understanding the biophysical constraints is a key to model the molecular evolution of proteins (*Echave and Wilke, 2017*; *Luksza and Lässig, 2014*; *Nourmohammad et al., 2013*; *Otwinowski et al., 2018*; *Mustonen et al., 2008*; *Wylie and Shakhnovich, 2011*; *Lässig et al., 2017*). Through a systematic analysis of pairwise epistasis, this study shows that the local net charge is a major biophysical molecular phenotype that constrains the evolution of an antigenic region in human influenza H3N2 NA. An important feature for this antigenic region of interest is that pairwise epistasis is highly conserved across diverse genetic backgrounds, which in turn is correlated with residue coevolution in naturally circulating human influenza H3N2 virus. Although the homologous regions in the H1N1 NA and influenza B NA do not evolve as extensively as human influenza H3N2 NA (*Figure 5—figure supplement 6*), residue 329 coevolves with 344 in seasonal influenza H1N1 NA, indicating that similar local net charge balancing may also influence the evolution of H1N1 NA. In fact, earlier theoretical studies have indicated that charge balancing is one of the strongest signatures of correlated evolution across different protein families (*Neher, 1994*; *Magliery and Regan, 2004*; *Callahan et al., 2011*). Our study here further provided empirical evidence for this phenomenon.

A key finding of this study is that the optimal local net charge at the antigenic region of interest is slightly negative, and an increase or decrease in local net charge is deleterious. This characteristic suggests that the local net charge may be the key molecular phenotype under stabilizing and balancing selection, imposing a biophysical constraint on evolution of the NA antigenic region. This local net charge may have several nonexclusive functional roles. First, since the host cell membrane and sialylated glycan receptor are both negatively charged, charge distribution on the virus surface, including the antigenic region of interest, may affect the kinetics of host membrane attachment and virus release. Second, the antigenic region of interest is proximal to the catalytic active site (*Figure 1a*), the local net charge may affect the catalytic efficiency of NA. Lastly, the local net charge may influence the protein stability of NA (*de Graff et al., 2016*; *Raghunathan et al., 2013*). Understanding the detailed molecular mechanisms of biophysical constraints, albeit beyond the scope of this study, will likely further enhance the ability to model evolution.

One interesting observation in this study is that the ancestral strain HK68 has a much lower mutational tolerance at the antigenic region of interest than the subsequent strains, although the topology of the local fitness landscapes is largely conserved across strains. This result suggests that other biophysical features of NA are evolving over time and that they epistatically interact with the antigenic region of interest in a residue nonspecific manner. Nonspecific epistasis is often related to protein

stability (*Starr and Thornton, 2016*), which is best described by the threshold robustness model (*Bloom et al., 2006*). Under the threshold robustness model, slightly destabilizing mutations may have neutral fitness when the protein has an excess stability margin but are deleterious when the protein is marginally stable. Threshold robustness model can be used to explain the differential variant fitness distribution among the six genetic backgrounds in our study. For example, it is possible that Mos99 NA has an excess stability margin such that many variants are nearly neutral despite being slightly destabilizing. In contrast, the same slightly destabilizing variants are highly deleterious in HK68 NA because it is marginally stable. When comparing the variant fitness in different genetic backgrounds, some nonlinearity can be observed (*Figure 2b*), which is a feature of the threshold robustness model (*Otwinowski et al., 2018*; *Bloom Jesse et al., 2005*). Nevertheless, additional studies are needed to confirm whether the difference in variant fitness distribution among genetic backgrounds is due to protein stability or other biophysical factors.

Predicting the evolution of human influenza virus is a challenging task, yet important for seasonal influenza vaccine development. An accurate predictive model of human influenza evolution likely requires an integration of epitope information (*Luksza and Lässig, 2014*), antigenic data (*Neher et al., 2016*), mutant fitness measurement (*Lee et al., 2018*), and experimental selection for antibody escape variants (*Li et al., 2016*). This work further suggests that a biophysical epistatic model of antigenic fitness landscape can also be instrumental in modeling the evolution of human influenza virus. As the knowledge about the evolutionary biology of influenza virus accumulates, a unifying model that can accurately predict emerging mutation may one day be built.

# Materials and methods

**Key resources table**

| Reagent type (species) or resource | Designation | Source or reference | Identifiers | Additional information |
|---|---|---|---|---|
| Cell line (*Homo sapiens*) | MDCK-SIAT1 | Sigma-Aldrich | Cat#: 05071502-1VL | |
| Cell line (*H. sapiens*) | HEK 293T | Scripps Research | N/A | |
| Strain, strain background (*Escherichia coli*) | MegaX DH10B T1R | Thermo Fisher Scientific | Cat#: C640003 | |
| Recombinant DNA reagent | A/WSN/33 (H1N1) eight-plasmid reverse genetics system | *Neumann et al., 1999* | N/A | |
| Commercial assay or kit | Lipofectamine 2000 | Thermo Fisher Scientific | Cat#: 11668019 | |
| Commercial assay or kit | KOD Hot Start DNA Polymerase | EMD Millipore | Cat#: 71086-3 | |
| Software, algorithm | Python | Python Software Foundation | RRID:SCR_008394 | |
| Software, algorithm | R | R Core Team | RRID:SCR_001905 | |

## Cell lines

HEK 293T (human embryonic kidney) cells and MDCK-SIAT1 (Madin–Darby canine kidney) cells were used in this study. The identification of the cell lines was confirmed morphologically. Cells were maintained in a humidified 37°C, 5% $CO_2$ incubator and cultured in Dulbecco's modified Eagle's medium (DMEM) (Life Technologies), supplemented with 10% fetal bovine serum (FBS) (VWR), and 1% penicillin-streptomycin (Life Technologies). Cells were tested monthly for mycoplasma contamination. Mycoplasma contamination was not detected.

## Recombinant influenza virus

All H3N2 viruses generated in this study were based on the influenza A/WSN/33 (H1N1) eight-plasmid reverse genetics system (*Neumann et al., 1999*). Chimeric 6:2 reassortments were employed with the HA ectodomains from the H3N2 A/Hong Kong/1/1968 (HK68) and the entire NA coding region from the strains of interest (*Wu et al., 2017*). For HA, the ectodomain was from HK68, whereas

the noncoding region, N-terminal secretion signal, C-terminal transmembrane domain, and cytoplasmic tail were from A/WSN/33. For NA, the entire coding region was from the strains of interest, whereas the noncoding region of NA was from A/WSN/33. H3N2 strains of interest in this study were as follows with GISAID (*Shu and McCauley, 2017*) accession numbers in parentheses: A/Hong Kong/1/1968 (EPI_ISL_245769), A/Bangkok/1/1979 (EPI_ISL_122020), A/Beijing/353/1989 (EPI_ISL_123212), A/Moscow/10/1999 (EPI_ISL_127595), A/Victoria/361/2011 (EPI_ISL_158723), and A/Hong Kong/2671/2019 (EPI_ISL_882915). For virus rescue using the eight-plasmid reverse genetics system, transfection was performed in HEK 293T/MDCK-SIAT1 cells (Sigma-Aldrich, Cat#: 05071502-1VL) that were co-cultured (ratio of 6:1) at 60% confluence using Lipofectamine 2000 (Life Technologies) according to the manufacturer's instructions. At 24 hr post-transfection, cells were washed twice with phosphate-buffered saline (PBS) and cell culture medium was replaced with OPTI-MEM medium supplemented with 0.8 µg mL$^{-1}$ TPCK-trypsin. Virus was harvested at 72 hr post-transfection. For measuring virus titer by TCID$_{50}$ assay, MDCK-SIAT1 cells were washed twice with PBS prior to the addition of virus, and OPTI-MEM medium was supplemented with 0.8 µg mL$^{-1}$ TPCK-trypsin.

## Mutant library construction

For each mutant library, insert and vector fragments were generated by PCR using PrimeSTAR Max DNA Polymerase (Takara) according to the manufacturer's instructions, with WT NA-encoding plasmid (pHW2000-NA) as templates. Primers for the insert contained the combinatorial mutations of interest and are shown in *Supplementary file 1*. For insert fragment, two rounds of PCR were performed. Forward primers (P2 set) and reverse primers (P3 set) were mixed at the indicated molar ratio and used for the first round PCR (*Supplementary file 1*). Products from the first round PCR were then purified using Monarch DNA Gel Extraction Kit (New England Biolabs) and used as the templates for the second round insert PCR. Forward primers (P1 set) and reverse primers (P3 set) were mixed at the indicated molar ratio and used for the second round PCR (*Supplementary file 1*). Of note, the same reverse primers (P3 set) were used for both rounds PCR. Primers for the vector PCR are also shown in *Supplementary file 1*. The final PCR products of the inserts and vectors were purified by PureLink PCR purification kit (Thermo Fisher Scientific), digested by DpnI and BsmBI (New England Biolabs), and ligated using T4 DNA ligase (New England Biolabs). The ligated product was transformed into MegaX DH10B T1R cells (Thermo Fisher Scientific). At least 1 million colonies were collected for each mutant library. Plasmid mutant libraries were purified from the bacteria colonies using Plasmid Midi Kit (QIAGEN).

## High-throughput fitness measurement of NA mutants

Each plasmid mutant library was rescued as described above (see section 'Recombinant influenza virus') in a T75 flask, titered by TCID$_{50}$ assay using MDCK-SIAT1 cells then stored at –80°C until use. To passage the virus mutant libraries, MDCK-SIAT1 cells in s T75 flask were washed twice with PBS and then infected with a multiplicity of infection (MOI) of 0.02 in OPTI-MEM medium containing 0.8 µg mL$^{-1}$ TPCK-trypsin. Infected cells were washed twice with PBS at 2 hr post-infection, then fresh OPTI-MEM medium containing 0.8 µg mL$^{-1}$ TPCK-trypsin was added to the cells. At 24 hr post-infection, supernatant containing the virus was collected. Viral RNA was extracted using QIAamp Viral RNA Mini Kit (QIAGEN). Purified viral RNA was reverse transcribed to cDNA using Superscript III reverse transcriptase (Thermo Fisher Scientific). The adapter sequence for Illumina sequencing was added to the plasmid or cDNA from the post-passaging virus mutant libraries by PCR using sequencing library preparation primers listed in *Supplementary file 1*. An additional PCR was performed to add the rest of the adapter sequence and index to the amplicon using primers: 5'-AAT GAT ACG GCG ACC ACC GAG ATC TAC ACT CTT TCC CTA CAC GAC GCT-3' and 5'-CAA GCA GAA GAC GGC ATA CGA GAT XXX XXX GTG ACT GGA GTT CAG ACG TGT GCT-3'. Positions annotated by an X represent the nucleotides for the index sequence. The final PCR products were purified by PureLink PCR purification kit (Thermo Fisher Scientific) and submitted for next-generation sequencing using Illumina MiSeq PE250.

## Sequencing data analysis

Sequencing data were obtained in FASTQ format and analyzed using custom Python scripts. Briefly, sequences were parsed by SeqIO module in BioPython (*Cock et al., 2009*). After trimming the primer

sequences, both forward and reverse-complement of the reverse reads were translated into protein sequences. A paired-end read was then filtered and removed if the protein sequences of the forward and reverse-complement of the reverse reads did not match. Subsequently, amino acids at the residues of interest were extracted. The number of reads corresponding to each of the 864 variants was then counted. The unnormalized fitness of each variant in each replicate was estimated as follows:

$$f_i = log_{10} \frac{(Output_{Count i}+1)/(input_{Count i}+1)}{(Output_{Count WT}+1)/(input_{Count WT}+1)}$$

where the $Output_{Count i}$ represents the number of reads corresponding to variant in the post-passaging virus mutant library, and the $input_{Count i}$ represents the number of reads corresponding to variant in the plasmid mutant library. A pseudocount of 1 was added to the counts to avoid division by zero. Of note, the WT sequence of Vic11 contains a naturally rare variant T329. As a result, the WT sequence of Vic11 was not included in our mutant library design. However, due to incomplete DpnI digestion of the vector during mutant library construction, the WT sequence of Vic11 was present in the Vic11 mutant library and could be detected in the next-generation sequencing data.

The final fitness value for each mutant is

$$f_i = log_{10} \left( \frac{(Output_{Count i,rep1}+1)/(input_{Count i,rep1}+1)}{(Output_{Count WT,rep1}+1)/(input_{Count WT,rep1}+1)} + \frac{(Output_{Count i,rep2}+1)/(input_{Count i,rep2}+1)}{(Output_{Count WT,rep2}+1)/(input_{Count WT,rep2}+1)} \right)$$

where *rep1* and *rep2* represent replicate 1 and replicate 2, respectively. The final fitness value of each variant is listed in *Figure 2—source data 1*.

## Total charge of residues of interest

Positively charged amino acids (K/R) were assigned with a charge of 1, negatively charged amino acids (D/E) were –1, and neutral amino acids were 0. The net charge of a given variant is defined as the algebraic sum of the charges of the seven residues of interest. For example, the net charge of KNEGKKL, which has three positively charged amino acids and one negatively charged amino acid, is 2.

## Modeling fitness and decomposition of interactions

We modeled variant fitness as a nonlinear function of the sum of the additive and the pairwise effects by MAVE-NN (*Tareen et al., 2020*). Statistical learning model was trained to estimate the latent phenotype $\phi$ and prediction fitness $y$.

The sum of the additive and the pairwise effects was defined as a latent phenotype $\phi_{pairwise}$,

$$\phi_{pairwise} \left( \vec{x}; \vec{\theta} \right) = \theta_0 + \sum_{l=0}^{L-1} \sum_c \theta_{l: c} x_{l: c} + \sum_{l=0}^{L-2} \sum_{l'=l+1}^{L-1} \sum_{c,c'} \theta_{l: c,l': c'} x_{l: c} x_{l': c'}$$

where $L$ is the length of the sequences, $C$ is the total number of amino acids, and

$$x_{l: c} = \begin{cases} 1 \ if \ character \ c \ occurs \ at \ position l \\ 0 \ otherwise, \end{cases}$$ is the one-hot encoding of the sequence at position $l$

when amino acid is $c$. $\vec{\theta}$ represents the weight of the additive and the pairwise effects. Fitness is then modeled as a nonlinear function of the sum of tanh,

$$\widehat{y} = g \left( \phi; \vec{\alpha} \right) = a + \sum_{k=0}^{K-1} b_k tanh \left( c_k \phi + d_k \right)$$

where $K$ specifies the number of 'hidden nodes' contributing to the sum, and $\vec{\alpha} = \{a, b_k, c_k, d_k\}$ are the trainable parameters.

To train the above model, dataset was reformatted to a set of $N$ observations, $\left\{ \left( \vec{x}_n, y_n \right) \right\}_{n=1}^N$, where each observation comprises sequence $\vec{x}_n$ and its fitness $y_n$. Then, the dataset was randomly divided into a training set, a validation set, and a test set with a ratio of 0.64:0.16:0.2. Model was evaluated using repeated k-fold cross-validation, and hyperparameters were chosen by maximizing the $R^2$ of model prediction and Pearson correlation coefficient of model parameters (*Otwinowski and Nemenman, 2013*). Notably, we find that the $R^2$ of our model prediction is insensitive to regularization and largely depends on the quality of the variant fitness data (*Figure 3—figure supplement 1*).

## Estimating epistasis between charged states

Amino acids are classified according to charges. Positive (+) represents positively charged amino acids (K/R), negative (-) represents negatively charged amino acids (D/E), and neutral (n) represents the remaining amino acids. All variants of the NA antigen region are then converted into charge states, named positive, negative, and neutral for each residue. For example, –344 represents K344 and R344. The epistasis value of a given charge state is the average over the epistasis values of all amino acid variants in the specified charge state.

## Analysis of natural sequences

A total of 66,562 full-length NA protein sequences from human H3N2 were downloaded from the GISAID (http://gisaid.org) (**Shu and McCauley, 2017**; **Supplementary file 2**, **Figure 1—source data 1**). Amino acid sequences of NA residues 328, 329, 344, 367, 368, 369, and 370 in individual strains were extracted. Individual sequences were grouped by the year of isolation, and their mean fitness in different genetic backgrounds is plotted in **Figure 2c**. The human H3N2 NA protein sequences used in this study are listed in **Figure 1—source data 1**. The same analysis was performed on the NA homologous region of influenza A H1N1 and influenza B.

## Inference of natural coevolution score

The change in the natural frequency of the charge states (+/-/n) at residue from year $n-1$ to year $n$ was computed as

$$\Delta f\left(s_i, n\right) = \sum_{aa_i \in s_i} f_{aa_i, n} - \sum_{aa_i \in s_i} f_{aa_i, n-1}$$

where $s_i$ is the charge states (+/-/n) at residue , and $f_{aa_i, n}$ is the natural frequency of the amino acid variant (*aa*) at residue in year $n$. Charge state '+' included amino acids K and R. Charge state '-' included amino acids D and E. Charge state 'n' included the remaining amino acids.

All local maxima were then selected and defined as peak of frequency change using find_peaks module in SciPy. Subsequently, to evaluate the proximity of peaks among two charge states, a coevolution score was calculated as the sum of exponential weights of all pairwise peak distances from a given mutation pair $(i,j)$ :

$$E\left(i,j\right) = \sum_k^N Ce^{-d_k}$$

where $d_k$ is the time separation (in years) between two peaks and $C$ is initial value. $C = 1$ if both peaks have the same sign, otherwise $C = -1$ (i.e., one peak is positive and the other is negative). Pairs of peaks with a time separation $d_k$ of longer than 5 years were discarded. See **Figure 5—figure supplement 3** for a schematic overview of calculating the coevolution score.

## Construction of evolutionary trajectories based on the fitness data

To construct an ensemble of evolutionary trajectories using the fitness data, we assume that the same haplotype would not reappear along an evolutionary trajectory. In addition, only those seven-residue variants in our mutant libraries that had a net charge of –1, 0, or 1 were included. Besides, we only included those amino acid variants that were in our mutant libraries. An ensemble of evolutionary trajectories was constructed for all six strains of interest using their corresponding WT sequences as starting points. The number of steps in the constructed trajectory is equivalent to the number of mutations that naturally emerged following the isolation of the focal strain and prior to the point when the subsequent strains of interest were sampled. For example, there were three mutations that naturally merged between 1989 and 1999. As a result, three steps were included for the constructed evolutionary trajectory of Bei89. The fitness value at each step in the evolutionary trajectory was equivalent to that of the corresponding mutant, with the fitness of the WT set as 1 (see section 'Sequencing data analysis').

## Code availability

Custom Python scripts for analyzing the fitness landscape data have been deposited to https://github.com/Wangyiquan95/NA_EPI (**Wu, 2021**; copy archived at swh:1:rev:2126f527add1c02cd9490a520d647b0700a18a2c).

## Acknowledgements

This work was supported by DFG grant (SFB1310) for Predictability in Evolution (AN), MPRG funding through the Max Planck Society (AN), the Royalty Research Fund from the University of Washington (AN), National Institutes of Health (NIH) R35 GM142795 (AN), R00 AI139445 (NCW) and DP2 AT011966 (NCW). We thank Justin Kinney and Ammar Tareen for helpful discussion, and the Roy J Carver Biotechnology Center at the University of Illinois at Urbana-Champaign for assistance with next-generation sequencing.

## Additional information

### Competing interests

Armita Nourmohammad: Reviewing editor, eLife. The other authors declare that no competing interests exist.

### Funding

| Funder | Grant reference number | Author |
| --- | --- | --- |
| Deutsche Forschungsgemeinschaft | SFB1310 | Armita Nourmohammad |
| Max Planck Society | MPRG funding | Armita Nourmohammad |
| University of Washington | Royalty Research Fund: A153352 | Armita Nourmohammad |
| National Institutes of Health | R00 AI139445 | Nicholas C Wu |
| National Institutes of Health | DP2 AT011966 | Nicholas C Wu |
| National Institutes of Health | R35 GM142795 | Armita Nourmohammad |

The funders had no role in study design, data collection and interpretation, or the decision to submit the work for publication.

### Author contributions

Yiquan Wang, Conceptualization, Formal analysis, Investigation, Methodology, Writing - original draft, Writing – review and editing; Ruipeng Lei, Conceptualization, Investigation, Methodology, Writing - original draft, Writing – review and editing; Armita Nourmohammad, Formal analysis, Investigation, Methodology, Writing – review and editing; Nicholas C Wu, Conceptualization, Formal analysis, Funding acquisition, Investigation, Methodology, Supervision, Writing - original draft, Writing – review and editing

### Author ORCIDs

Yiquan Wang  http://orcid.org/0000-0002-1954-9808
Ruipeng Lei  http://orcid.org/0000-0002-4652-3400
Armita Nourmohammad  http://orcid.org/0000-0002-6245-3553
Nicholas C Wu  http://orcid.org/0000-0002-9078-6697

### Decision letter and Author response

Decision letter https://doi.org/10.7554/eLife.72516.sa1
Author response https://doi.org/10.7554/eLife.72516.sa2

## Additional files

### Supplementary files
• Supplementary file 1. Primer sequences.

- Supplementary file 2. Number of human H3N2 neuraminidase (NA) sequences per year.
- Transparent reporting form

## Data availability

Raw sequencing data have been submitted to the NIH Short Read Archive under accession number: BioProject PRJNA742436. Custom python scripts for analyzing the deep mutational scanning data have been deposited to https://github.com/Wangyiquan95/NA_EPI, (copy archived at https://archive.softwareheritage.org/swh:1:rev:2126f527add1c02cd9490a520d647b0700a18a2c).

The following dataset was generated:

| Author(s) | Year | Dataset title | Dataset URL | Database and Identifier |
|-----------|------|---------------|-------------|-------------------------|
| Wang Y, Nc Wu | 2021 | Deep mutational scanning of human H3N2 influenza virus NA residues 328, 329, 344, 367, 368, 369, and 370 | https://www.ncbi.nlm.nih.gov/bioproject/742436 | NIH Short Read Archive BioProject, PRJNA742436 |

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
