## [Editor Report]

This paper presents a systematic analysis of the fitness landscape of the influenza virus protein neuraminidase (NA). The authors generate 864 different combinations of amino acids at seven positions in six genetic backgrounds sampled 10 years apart and measure the fitness of the resulting virus. This fitness landscape is characterized by strong epistatic interactions, including a strong tendency to maintain the local charge of the protein. Such systematic characterizations of important proteins of viral pathogens are crucial to develop principled models to understand and predict their evolution.

---

## [Decision Letter]

**Decision letter after peer review:**

Thank you for submitting your article "Antigenic evolution of human influenza H3N2 neuraminidase is constrained by charge balancing" for consideration by *eLife*. Your article has been reviewed by 3 peer reviewers, including Richard A Neher as Reviewing Editor and Reviewer #1, and the evaluation has been overseen by Betty Diamond as the Senior Editor. The following individuals involved in review of your submission have agreed to reveal their identity: Christopher J R Illingworth (Reviewer #2); Claus O Wilke (Reviewer #3).

Essential revisions:

1) The primary conclusion from your work is that evolution of NA is heavily constraint by charge balancing and that this constraint could potentially be used to predict evolution. It would be great if this could be quantified. How does the realized trajectory compare to an ensemble of possible ones? How many of the past substitutions could have been predicted? Are similar constraints seen in homologous regions of A/H1N1 of B? Some additional exploration of this would strengthen the paper.

2) The structural interpretation of the results could potentially be strengthened by analyzing side-chain positions rather than C-α positions. Comparisons of structures with different local charge might also be informative.

3) The variation among the estimates of additive effects is surprising. Is this variation smaller between similar backgrounds (10 years apart) compared to distant backgrounds? How well does an "additive only" model fit the data and how variable are the inferred effects? Some additional discussion of this would help.

4) The experiments presented here are different from most DMS experiments: All possible combinations of a subset of mutations are analyzed, rather than all possible point mutations. It would be good to highlight this difference and possibly use a different term.

The reviews below contain a number of other suggestions that we encourage you to consider.

*Reviewer #1 (Recommendations for the authors):*

– I have trouble interpreting the result that additive fitness effects are all over the place when comparing backgrounds, while pairwise effects are conserved. How much does this depend on the model choice? You could also use un-ambiguous Fourier transformation on subsets of binary hypercubes. And fit a model with only addititive coefficients. How much would the coefficients vary in that case? How much less variance is captured? Is there a sense that additive fitness effects are more strongly correlated with comparing backgrounds 10y appart then 20, 30, 40y apart?

– The part on analyzing natural variation in charge and prediction could be expanded and explored a little more. Net charge of the regions tends to be either -1,0, or +1, but it is not apriori clear that the realized trajectory is much more narrow in its charge distribution than random permutations of the mutations of the last 50 years are. From the experimental data, a charge of -2 seems better than +1, but +1 has been dominant in the last few years, while -2 is only sporadically observed. There is of course only one realization with a long correlation time. To strengthen the case for predictive power, would it be feasible to repeat this analysis on homologous residues in H1N1 (and maybe B and the H1N1pdm)?

*Reviewer #2 (Recommendations for the authors):*

1. The authors note that the region studied is subject to antigenic evolution. To what extent do the authors believe that charge balancing imposes a negative constraint upon evolution, as opposed to epistasis driving compensating changes in NA under positive selection following an immunologically driven change in the virus that coincidentally impacts residue charge? This is not to nullify the finding, but of the substiutions observed in this region over time, how many could have been predicted in advance on the basis of fitness landscapes derived here (accounting for their charge or non-charge effects)? Can the authors predict likely changes in this region that are likely to be observed next?

2. With this in mind, I found Figure 5a hard to understand or interpret. Would it be possible to show the data in a way that highlights the frequency of haplotypes by charge against time? I would find this easier to follow.

3. A small amount of work has been done looking at the consequences of charge for the local structure, though sidechain positions might be a little stochastic between crystal structures. Are there any consistent changes in structure arising from changes in local charge? That is, if multiple structures with different charge distributions are locally aligned (e.g. using VMD to perform local structural alignment on backbone atoms and RMSD to assess differences), are any changes in local conformation evident?

4. I am perhaps not sufficiently familiar with the terminology to understand what exactly was done in the virus rescue experiment. Could more details be provided?

5. Is it possible, potentially via a more superficial analysis, to suggest whether the region of charge conservation in NA might be preserved beyond the region under study? I am happy if the authors don't want to go down this route, though I wonder whether a figure along the lines of a revised Figure 5A would show possible routes for future exploration.

*Reviewer #3 (Recommendations for the authors):*

I think you need to be absolutely clear about whether you performed six experiments of 864 variants or 864 deep mutational scanning experiments of every possible mutation in NA. I think you did the former, but in many places the manuscript reads as if you did the latter. If you did six experiments of 864 variants, I believe you shouldn't use the term "deep mutational scanning" at all.

Neuraminidase has almost 400 residues, so a deep mutational scan of one NA variation would require almost 8000 mutations, ten times more than one of your experiment did (assuming you did only six experiments).

I also would like to emphasize that one of the main strong points of your study (if I understand correctly what you did) is that you systematically explored all possible combinations of a set of mutations. This is very different from deep mutational scanning, which usually only looks at single mutations and maybe sometimes at mutation pairs. By emphasizing deep mutational scanning, you are drawing attention away from this aspect of your study, even though that's the primary selling point of your study in my opinion.

Additionally, redoing the analysis of Figure 4 with side-chain distances should be straightforward. You could use either the smallest distance among all heavy atoms or the distance between the geometric center of the sidechain atoms, whichever is easier for you to calculate. Both should give you approximately the same results.

[Editors’ note: further revisions were suggested prior to acceptance, as described below.]

Thank you for resubmitting your work entitled "Antigenic evolution of human influenza H3N2 neuraminidase is constrained by charge balancing" for further consideration by *eLife*. Your revised article has been evaluated by Betty Diamond (Senior Editor) and a Reviewing Editor.

The manuscript has been improved but there are some remaining issues that need to be addressed, as outlined below:

Thank you very much for submitting the revision. I still think this study is elegant and important (as did my co-reviewers). Unfortunately, the additional analyses you have done did not provide strong support for the claim that charge conversation can be used to predict influenza evolution, while the central claim of charge being an important constraint remains convincing. I therefore think some parts of the manuscript need to be toned down. The last two sentences of abstract, for example, seem too strong:

"In addition, we show that residue coevolution in this antigenic region can be predicted by charge states and pairwise epistasis. Overall, this study demonstrates the importance of quantifying epistasis and the underlying biophysical constraints for building a predictive model of influenza evolution."

The main evidence that charge constrains natural evolution is the concordance of co-evolution scores and pairwise epistasis. I think it should be better explained what can and what can not be concluded from this and in what sense this is predictive.

The definition of the score is also somewhat confusing and I think there are some problems around lines 470 and 471. Why is s_i_ summed over when s_i_ is an argument to \delta f? The text below doesn't help to resolve the matter.

It would be nice if you made it a bit easier for the reader to piece together what exactly happened in natural H3N2 populations in the past 50y. Figure 1c shows the trajectories, but it is at times tough to link colors with amino acids and the associated changes in charge. Highlighting which events change charge would be helpful. These events underlie Figure 5a and ultimately Figure 5c. It should be possible to show more explicitly how these are connected, maybe by combining only charge changing frequency trajectories into one graph or by increasing panel 5a and annotating the curves with the underlying changes in genotype.

---

## [Author Response]

Essential revisions:1) The primary conclusion from your work is that evolution of NA is heavily constraint by charge balancing and that this constraint could potentially be used to predict evolution. It would be great if this could be quantified. How does the realized trajectory compare to an ensemble of possible ones? How many of the past substitutions could have been predicted? Are similar constraints seen in homologous regions of A/H1N1 of B? Some additional exploration of this would strengthen the paper.

Thank you for the comment. In the revised manuscript, we constructed an assemble of evolutionary trajectories using the fitness data, as described in a newly added subsection “Construction of evolutionary trajectories based on the fitness data” in the Methods section:

“To construct an assemble of evolutionary trajectories using the fitness data, we assume that the same haplotype would not reappear along an evolutionary trajectory. In addition, only those seven-residue variants in our mutant libraries that had a net charge of -1, 0, or 1 were included. Besides, we only included those amino acid variants that were in our mutant libraries. An assemble of evolutionary trajectories was constructed for all six strains of interest, using their corresponding WT sequences as starting points. The number of steps in the constructed trajectory is equivalent to the number of mutations that naturally emerged following the isolation of the focal strain and prior to the point when the subsequent strains of interest were sampled. For example, there were three mutations that naturally merged between 1989 and 1999. As a result, three steps were included for the constructed evolutionary trajectory of Bei89. The fitness value at each step in the evolutionary trajectory was equivalent to that of the corresponding mutant, with the fitness of the WT set as 1 (see “Sequencing data analysis” above).”

The results of this analysis are shown in Figure 5—figure supplement 6 and described in the Results section:

“We also attempted to use the fitness data to predict the identity of amino acid mutations along the evolutionary trajectory. Nonetheless, the realized evolutionary trajectories in nature are seldom the most probable ones among an ensemble of trajectories that were constructed from the fitness data (Figure 5—figure supplement 6). As a result, while our analysis demonstrates that a biophysically grounded epistatic landscape is informative for the prediction of the natural coevolution of residues in this NA antigenic region, additional considerations are needed for the prediction of the exact amino acid mutations along the evolutionary trajectory.”

Part of the results are also described in the legend of Figure 5—figure supplement 6:

“Evolutionary trajectories were constructed based on the fitness data (see Methods). The realized trajectory in nature is highlighted in red. This result demonstrates that natural evolutionary trajectory does not always maximize the replication fitness. This discrepancy is likely due to other selection pressures in nature (e.g. immune selection pressure) that were not being captured in our in vitro fitness measurement, or due to changes in the genetic background through epistatic interactions that were not captured by our experiments.”

We also explored whether the charge balancing constraint seen in H3N2 NA can also be observed in the NA homologous region of H1N1 and influenza B. Figure 5—figure supplement 7 in the revised manuscript analyzes the evolution of the NA homologous region in seasonal H1N1 virus, 2009 pandemic H1N1 Influenza A virus, and influenza B virus. The observations are discussed in the Discussion section:

“Although the homologous regions in the H1N1 NA and influenza B NA do not evolve as extensively as human influenza H3N2 NA (Figure 5—figure supplement 7), residue 329 coevolves with 344 in seasonal influenza H1N1 NA, indicating that similar local net charge balancing may also influence the evolution of H1N1 NA and influenza B NA.”

2) The structural interpretation of the results could potentially be strengthened by analyzing side-chain positions rather than C-α positions. Comparisons of structures with different local charge might also be informative.

Thank you for the suggestion. In the revised manuscript, we replaced the Cα analysis in Figure 4d-f and Figure 4—figure supplement 3 by a side-chain analysis to examine the relationship between pairwise epistasis and the side-chain distance. The result of this analysis is described in the Results section:

“Consistently, pairwise epistasis between two amino acid variants does not correlate with their side-chain – side-chain distances (Figure 4d-f and Figure 4—figure supplement 3), further substantiating that direct side-chain – side-chain interaction is not a determinant for epistasis here.”

As indicated in the legend of Figure 4—figure supplement 3:

“The side-chain – side-chain distance was computed based on the geometric center of the sidechain atoms from each residue of interest^66^.”

3) The variation among the estimates of additive effects is surprising. Is this variation smaller between similar backgrounds (10 years apart) compared to distant backgrounds? How well does an "additive only" model fit the data and how variable are the inferred effects? Some additional discussion of this would help.

In the Results section of the revised manuscript, we have made a comment about the relationship between variation of additive fitness contributions and similarity of genetic backgrounds:

“Of note, the variation of additive fitness contributions across genetic backgrounds does not seem to strictly depend on the similarity between genetic backgrounds, since the correlation between the additive fitness contributions of HK68 and HK19 (Pearson correlation = 0.46) is much higher than that of HK68 and Bk79 (Pearson correlation = 0.12), which have a shorter time separation. This observation points at a possibility for complex epistatic interactions between the antigenic region of interest and other regions on NA.”

In the revised manuscript, we added an analysis by fitting an “additive only” model to the data in Figure 3—figure supplement 4.

“Consistently, an “additive only” model, without accounting for epistasis, shows a poor fit to the fitness landscape data compared to the model above with epistasis (Figure 3—figure supplement 4a), despite the fact that the inferred additive fitness effects correlate well between the two models (Figure 3—figure supplement 4b).”

4) The experiments presented here are different from most DMS experiments: All possible combinations of a subset of mutations are analyzed, rather than all possible point mutations. It would be good to highlight this difference and possibly use a different term.

In the revised manuscript, we have avoided using deep mutational scanning to describe our experimental design. We have also highlighted the difference between conventional deep mutational scanning experiments and our approach in the introduction section:

“Unlike saturation mutagenesis in conventional deep mutational scanning, combinatorial mutagenesis enables us to measure the fitness of high-order mutants.”

We also mentioned the difference in the Results section:

“Unlike conventional deep mutational scanning, which studies all possible single amino acid mutations across a protein or domain of interest, our approach analyzes all possible combinations of a subset of mutations.”

The reviews below contain a number of other suggestions that we encourage you to consider.Reviewer #1 (Recommendations for the authors):– I have trouble interpreting the result that additive fitness effects are all over the place when comparing backgrounds, while pairwise effects are conserved. How much does this depend on the model choice? You could also use un-ambiguous Fourier transformation on subsets of binary hypercubes. And fit a model with only addititive coefficients. How much would the coefficients vary in that case? How much less variance is captured? Is there a sense that additive fitness effects are more strongly correlated with comparing backgrounds 10y appart then 20, 30, 40y apart?

Thank you for the suggestion. The difference in the additive effect is mainly reflected in the early strains, such as HK68 and BK79, while the difference is smaller for the strains in recent years. In the Results section of the revised manuscript, we have made a comment about the relationship between variation of additive fitness contributions and similarity of genetic backgrounds (see response to Essential Revisions above). We have also added an analysis by fitting an “additive only” model to the data in Figure 3—figure supplement 4 (see response to Essential Revisions above).

– The part on analyzing natural variation in charge and prediction could be expanded and explored a little more. Net charge of the regions tends to be either -1,0, or +1, but it is not apriori clear that the realized trajectory is much more narrow in its charge distribution than random permutations of the mutations of the last 50 years are. From the experimental data, a charge of -2 seems better than +1, but +1 has been dominant in the last few years, while -2 is only sporadically observed. There is of course only one realization with a long correlation time. To strengthen the case for predictive power, would it be feasible to repeat this analysis on homologous residues in H1N1 (and maybe B and the H1N1pdm)?

As a suggested by the reviewer, we explored whether such a charge constrain is conserve in the NA homologous region of A/H1N1 and influenza B. The result is presented as Figure 5—figure supplement 7 in the revised manuscript and described in the Discussion section (see response to Essential Revisions above).

Reviewer #2 (Recommendations for the authors):1. The authors note that the region studied is subject to antigenic evolution. To what extent do the authors believe that charge balancing imposes a negative constraint upon evolution, as opposed to epistasis driving compensating changes in NA under positive selection following an immunologically driven change in the virus that coincidentally impacts residue charge? This is not to nullify the finding, but of the substiutions observed in this region over time, how many could have been predicted in advance on the basis of fitness landscapes derived here (accounting for their charge or non-charge effects)? Can the authors predict likely changes in this region that are likely to be observed next?

In the revised manuscript, we found that the realized evolutionary trajectories in nature are seldom the most probable ones among an ensemble of trajectories that were constructed from the fitness data (see response to Essential Revisions above). As a result, while our analysis demonstrates that a biophysically grounded epistatic landscape is informative for the prediction of the natural coevolution of residues in this NA antigenic region, additional considerations are needed for the prediction of the exact amino acid mutations along the evolutionary trajectory.

2. With this in mind, I found Figure 5a hard to understand or interpret. Would it be possible to show the data in a way that highlights the frequency of haplotypes by charge against time? I would find this easier to follow.

We have updated Figure 5a to improve the clarify of data presentation.

3. A small amount of work has been done looking at the consequences of charge for the local structure, though sidechain positions might be a little stochastic between crystal structures. Are there any consistent changes in structure arising from changes in local charge? That is, if multiple structures with different charge distributions are locally aligned (e.g. using VMD to perform local structural alignment on backbone atoms and RMSD to assess differences), are any changes in local conformation evident?

Unfortunately, there are only four human H3N2 NA structures available in the PDB. They are from strains A/Memphis/31/98 (Mem98), A/Perth/16/2009 (Perth09), A/Tanzania/205/2010 (Tan10), and A/Minnesota/11/2010 (Min10). While Tan10 has a net charge of -1 in the antigenic site of interest, the other three strains all have a net charge of 0. At the antigenic site of interest, the RMSD between Tan10 and other three strains with net charge of 0 ranges from 0.18 to 0.27 Å, whereas the RMSD between those three strains with net charge of 0 ranges from 0.15 to 0.19 Å. Although these differences are neglibible, we are not comfortable in drawing conclusions regarding the consequences of charge for the local structure due to the small samples size.

4. I am perhaps not sufficiently familiar with the terminology to understand what exactly was done in the virus rescue experiment. Could more details be provided?

In the revised manuscript, details of the virus rescue experiment are described in the methods section under subsection “Recombinant influenza virus”:

“For virus rescue using the eight-plasmid reverse genetics system, transfection was performed in HEK 293T/MDCK-SIAT1 cells (Σ-Aldrich, catalog number: 050715021VL) that were co-cultured (ratio of 6:1) at 60% confluence using lipofactamine 2000 (Life Technologies) according to the manufacturer’s instructions. At 24 hours post-transfection, cells were washed twice with phosphate-buffered saline (PBS) and cell culture medium was replaced with OPTI-MEM medium supplemented with 0.8 μg ml^-1^ TPCK-trypsin. Virus was harvested at 72 hours post-transfection. For measuring virus titer by TCID_50_ assay, MDCK-SIAT1 cells were washed twice with PBS prior to the addition of virus, and OPTIMEM medium was supplemented with 0.8 μg ml^-1^ TPCK-trypsin.”

5. Is it possible, potentially via a more superficial analysis, to suggest whether the region of charge conservation in NA might be preserved beyond the region under study? I am happy if the authors don't want to go down this route, though I wonder whether a figure along the lines of a revised Figure 5A would show possible routes for future exploration.

This is a great suggestion. In fact, we are wrapping up another much more in-depth study that is related to charge conservation in another region of NA. As a result, we prefer not to analyze other regions in the current study.

Reviewer #3 (Recommendations for the authors):I think you need to be absolutely clear about whether you performed six experiments of 864 variants or 864 deep mutational scanning experiments of every possible mutation in NA. I think you did the former, but in many places the manuscript reads as if you did the latter. If you did six experiments of 864 variants, I believe you shouldn't use the term "deep mutational scanning" at all.Neuraminidase has almost 400 residues, so a deep mutational scan of one NA variation would require almost 8000 mutations, ten times more than one of your experiment did (assuming you did only six experiments).

In the revised manuscript, we have avoided using deep mutational scanning to describe our experimental design. Instead, we described our approach as “a high-throughput experimental approach that coupled combinatorial mutagenesis and next-generation sequencing”.

I also would like to emphasize that one of the main strong points of your study (if I understand correctly what you did) is that you systematically explored all possible combinations of a set of mutations. This is very different from deep mutational scanning, which usually only looks at single mutations and maybe sometimes at mutation pairs. By emphasizing deep mutational scanning, you are drawing attention away from this aspect of your study, even though that's the primary selling point of your study in my opinion.

Thank you for the comment. In the revised manuscript, we have highlighted the difference between conventional deep mutational scanning experiments and our approach (see response to Essential Revisions above).

Additionally, redoing the analysis of Figure 4 with side-chain distances should be straightforward. You could use either the smallest distance among all heavy atoms or the distance between the geometric center of the sidechain atoms, whichever is easier for you to calculate. Both should give you approximately the same results.

Thanks for the suggestion. Geometric center of the side-chain atoms was used in our revised analysis in Figure 4.

[Editors’ note: what follows is the authors’ response to the second round of review.]

Thank you very much for submitting the revision. I still think this study is elegant and important (as did my co-reviewers). Unfortunately, the additional analyses you have done did not provide strong support for the claim that charge conversation can be used to predict influenza evolution, while the central claim of charge being an important constraint remains convincing. I therefore think some parts of the manuscript need to be toned down. The last two sentences of abstract, for example, seem too strong:"In addition, we show that residue coevolution in this antigenic region can be predicted by charge states and pairwise epistasis. Overall, this study demonstrates the importance of quantifying epistasis and the underlying biophysical constraints for building a predictive model of influenza evolution."

In the revised manuscript, the last two sentences of abstract are toned down:

“In addition, we show that residue coevolution in this antigenic region is correlated with the pairwise epistasis between charge states. Overall, this study demonstrates the importance of quantifying epistasis and the underlying biophysical constraint for building a model of influenza evolution.”

We have also toned down other related sentences in the manuscript (see tracked changes).

The main evidence that charge constrains natural evolution is the concordance of co-evolution scores and pairwise epistasis. I think it should be better explained what can and what can not be concluded from this and in what sense this is predictive.

Edits are made in the revised manuscript to emphasize that our results show a correlation between epistasis and co-evolution, but without mentioning that our results have predictive power.

The definition of the score is also somewhat confusing and I think there are some problems around lines 470 and 471. Why is s_i summed over when s_i is an argument to \Δ f? The text below doesn't help to resolve the matter.

Thank you for catching this problem, which is now resolved in the revised manuscript with additional clarifications.

It would be nice if you made it a bit easier for the reader to piece together what exactly happened in natural H3N2 populations in the past 50y. Figure 1c shows the trajectories, but it is at times tough to link colors with amino acids and the associated changes in charge. Highlighting which events change charge would be helpful. These events underlie Figure 5a and ultimately Figure 5c. It should be possible to show more explicitly how these are connected, maybe by combining only charge changing frequency trajectories into one graph or by increasing panel 5a and annotating the curves with the underlying changes in genotype.

Thanks for the great suggestion. In the revised manuscript, labels are added to Figure 5a to indicate the key events. This is also described in the figure legend:

“Key mutation events that lead to changes in local net charge are indicated.”